# Fluctuation-dissipation relations for stochastic gradient descent

**Sho Yaida**
Facebook AI Research
Facebook Inc.
Menlo Park, California 94025, USA
`shoyaida@fb.com`

## Abstract

The notion of the stationary equilibrium ensemble has played a central role in statistical mechanics. In machine learning as well, training serves as generalized equilibration that drives the probability distribution of model parameters toward stationarity. Here, we derive stationary fluctuation-dissipation relations that link measurable quantities and hyperparameters in the stochastic gradient descent algorithm. These relations hold exactly for any stationary state and can in particular be used to adaptively set training schedule. We can further use the relations to efficiently extract information pertaining to a loss-function landscape such as the magnitudes of its Hessian and anharmonicity. Our claims are empirically verified.

## 1 Introduction

Equilibration rules the long-term fate of many macroscopic dynamical systems. For instance, as we pour water into a glass and let it be, the stationary state of tranquility is eventually attained. Zooming into the tranquil water with a microscope would reveal, however, a turmoil of stochastic fluctuations that maintain the apparent stationarity in balance. This is vividly exemplified by the Brownian motion (Brown, 1828): a pollen immersed in water is constantly bombarded by jittery molecular movements, resulting in the macroscopically observable diffusive motion of the solute. Out of the effort in bridging microscopic and macroscopic realms through the Brownian movement came a prototype of fluctuation-dissipation relations (Einstein, 1905; Von Smoluchowski, 1906). These relations quantitatively link degrees of noisy microscopic fluctuations to smooth macroscopic dissipative phenomena and have since been codified in the linear response theory for physical systems (Onsager, 1931; Green, 1954; Kubo, 1957), a cornerstone of statistical mechanics.

Machine learning begets another form of equilibration. As a model learns patterns in data, its performance first improves and then plateaus, again reaching apparent stationarity. This dynamical process naturally comes equipped with stochastic fluctuations as well: often given data too gigantic to consume at once, training proceeds in small batches and random selections of these mini-batches consequently give rise to the noisy dynamical excursion of the model parameters in the loss-function landscape, reminiscent of the Brownian motion. It is thus natural to wonder if there exist analogous fluctuation-dissipation relations that quantitatively link the noise in mini-batched data to the observable evolution of the model performance and that in turn facilitate the learning process.

Here, we derive such fluctuation-dissipation relations for the stochastic gradient descent algorithm. The only assumption made is stationarity of the probability distribution that governs the model parameters at sufficiently long time. Our results thus apply to generic cases with non-Gaussian mini-batch noises and nonconvex loss-function landscapes. Practically, the first relation (FDR1) offers the metric for assessing equilibration and yields an adaptive algorithm that sets learning-rate schedule on the fly. The second relation (FDR2) further helps us determine the properties of the loss-function landscape, including the strength of its Hessian and the degree of anharmonicity, i.e., the deviation from the idealized harmonic limit of a quadratic loss surface and a constant noise matrix.

Our approach should be contrasted with recent attempts to import the machinery of stochastic differential calculus into the study of the stochastic gradient descent algorithm (Mandt et al., 2015; Li et al., 2015; Mandt et al., 2017; Li et al., 2017; Smith & Le, 2018; Chaudhari & Soatto, 2017;

Jastrzebski et al., 2017; Zhu et al., 2018; An et al., 2018). This line of work all assumes Gaussian noises and sometimes additionally employs the quadratic harmonic approximation for loss-function landscapes. The more severe drawback, however, is the usage of the analogy with continuous-time stochastic differential equations, which is inconsistent in general (see Section 2.3.3). Instead, the stochastic gradient descent algorithm can be properly treated within the framework of the Kramers-Moyal expansion (Van Kampen, 1992; Gardiner, 2009; Risken, 1984; Radons et al., 1990; Leen & Moody, 1993).

The paper is organized as follows. In Section 2, after setting up notations and deriving a stationary fluctuation-dissipation theorem (FDT), we derive two specific fluctuation-dissipation relations. The first relation (FDR1) can be used to check stationarity and the second relation (FDR2) to delineate the shape of the loss-function landscape, as empirically borne out in Section 3. An adaptive scheduling method is proposed and tested in Section 3.3. We conclude in Section 4 with future outlooks.

## 2 FLUCTUATION-DISSIPATION RELATIONS

A model is parametrized by a weight coordinate, $\boldsymbol{\theta} = \{\theta_i\}_{i=1,\ldots,P}$. The training set of $N_s$ examples is utilized by the model to learn patterns in the data and the model's overall performance is evaluated by a full-batch loss function, $f(\boldsymbol{\theta}) \equiv \frac{1}{N_s} \sum_{\alpha=1}^{N_s} f_\alpha(\boldsymbol{\theta})$, with $f_\alpha(\boldsymbol{\theta})$ quantifying the performance of the model on a particular sample $\alpha$: the smaller the loss is, the better the model is expected to perform. The learning process can thus be cast as an optimization problem of minimizing the loss function. One of the most commonly used optimization schemes is the stochastic gradient descent (SGD) algorithm (Robbins & Monro, 1951) in which a mini-batch $\mathcal{B} \subset \{1, 2, \ldots, N_s\}$ of size $|\mathcal{B}|$ is stochastically chosen for training at each time step. Specifically, the update equation is given by

$$\boldsymbol{\theta}(t+1) = \boldsymbol{\theta}(t) - \eta \boldsymbol{\nabla} f^{\mathcal{B}}[\boldsymbol{\theta}(t)] , \tag{1}$$

where $\eta > 0$ is a learning rate and a mini-batch loss $f^{\mathcal{B}}(\boldsymbol{\theta}) \equiv \frac{1}{|\mathcal{B}|} \sum_{\alpha \in \mathcal{B}} f_\alpha(\boldsymbol{\theta})$. Note that

$$\left[\!\left[ \boldsymbol{\nabla} f^{\mathcal{B}}(\boldsymbol{\theta}) \right]\!\right]_{\mathrm{m.b.}} = \boldsymbol{\nabla} f(\boldsymbol{\theta}) , \tag{2}$$

with $\left[\!\left[ \ldots \right]\!\right]_{\mathrm{m.b.}}$ denoting the average over mini-batch realizations. For later purposes, it is convenient to define a full two-point noise matrix $\widetilde{C}$ through[1]

$$\widetilde{C}_{i,j}(\boldsymbol{\theta}) \equiv \left[\!\left[ \left[\partial_i f^{\mathcal{B}}(\boldsymbol{\theta})\right] \left[\partial_j f^{\mathcal{B}}(\boldsymbol{\theta})\right] \right]\!\right]_{\mathrm{m.b.}} \tag{3}$$

and, more generally, higher-point noise tensors

$$\widetilde{C}_{i_1,i_2,\ldots,i_k}(\boldsymbol{\theta}) \equiv \left[\!\left[ \left[\partial_{i_1} f^{\mathcal{B}}(\boldsymbol{\theta})\right] \left[\partial_{i_2} f^{\mathcal{B}}(\boldsymbol{\theta})\right] \cdots \left[\partial_{i_k} f^{\mathcal{B}}(\boldsymbol{\theta})\right] \right]\!\right]_{\mathrm{m.b.}} . \tag{4}$$

Below, we shall not make any assumptions on the distribution of the noise vector $\boldsymbol{\nabla} f^{\mathcal{B}}$ – other than that a mini-batch is independent and identically distributed from the $N_s$ training samples at each time step – and the noise distribution is therefore allowed to have nontrivial higher connected moments indicative of non-Gaussianity.

It is empirically often observed that the performance of the model plateaus after some training through SGD. It is thus natural to hypothesize the existence of a stationary-state distribution, $p_{\mathrm{ss}}(\boldsymbol{\theta})$, that dictates the SGD sampling at long time (see Section 2.3.4 for discussion on this assumption). For any *observable* quantity, $\mathcal{O}(\boldsymbol{\theta})$, – something that can be measured during training such as $\boldsymbol{\theta}^2$ and $f(\boldsymbol{\theta})$ – its stationary-state average is then defined as

$$\langle \mathcal{O}(\boldsymbol{\theta}) \rangle \equiv \int \mathrm{d}\boldsymbol{\theta} \, p_{\mathrm{ss}}(\boldsymbol{\theta}) \, \mathcal{O}(\boldsymbol{\theta}) . \tag{5}$$

---

[1] A connected noise covariant matrix, $C_{i,j}(\boldsymbol{\theta}) \equiv \widetilde{C}_{i,j}(\boldsymbol{\theta}) - [\partial_i f(\boldsymbol{\theta})][\partial_j f(\boldsymbol{\theta})]$, will not appear in fluctuation-dissipation relations below but scales nicely with mini-batch sizes as $\propto \frac{1}{|\mathcal{B}|}\left(1 - \frac{|\mathcal{B}|}{N_s}\right)$ (Li et al., 2017).

In general the probability distribution of the model parameters evolves as $p(\boldsymbol{\theta}, t+1) = [\![\int \mathrm{d}\boldsymbol{\theta}' \, p(\boldsymbol{\theta}', t)\delta\{\boldsymbol{\theta} - [\boldsymbol{\theta}' - \eta\boldsymbol{\nabla}f^{\mathcal{B}}(\boldsymbol{\theta}')]\}]\!]_{\mathrm{m.b.}}$ and in particular for the stationary state

$$
\begin{aligned}
\int \mathrm{d}\boldsymbol{\theta}\, p_{\mathrm{ss}}\left(\boldsymbol{\theta}, t\right) \mathcal{O}\left(\boldsymbol{\theta}\right) &= \int \mathrm{d}\boldsymbol{\theta}\, p_{\mathrm{ss}}\left(\boldsymbol{\theta}, t+1\right) \mathcal{O}\left(\boldsymbol{\theta}\right) \\
&= \left[\!\left[\int \mathrm{d}\boldsymbol{\theta} \int \mathrm{d}\boldsymbol{\theta}'\, p_{\mathrm{ss}}(\boldsymbol{\theta}', t)\delta\left\{\boldsymbol{\theta} - \left[\boldsymbol{\theta}' - \eta\boldsymbol{\nabla}f^{\mathcal{B}}\left(\boldsymbol{\theta}'\right)\right]\right\} \mathcal{O}\left(\boldsymbol{\theta}\right)\right]\!\right]_{\mathrm{m.b.}} \\
&= \int \mathrm{d}\boldsymbol{\theta}'\, p_{\mathrm{ss}}\left(\boldsymbol{\theta}'\right) \left[\!\left[\mathcal{O}\left[\boldsymbol{\theta}' - \eta\boldsymbol{\nabla}f^{\mathcal{B}}\left(\boldsymbol{\theta}'\right)\right]\right]\!\right]_{\mathrm{m.b.}}.
\end{aligned} \tag{6}
$$

Thus follows the master equation

$$
\langle \mathcal{O}\left(\boldsymbol{\theta}\right)\rangle = \left\langle \left[\!\left[\mathcal{O}\left[\boldsymbol{\theta} - \eta\boldsymbol{\nabla}f^{\mathcal{B}}\left(\boldsymbol{\theta}\right)\right]\right]\!\right]_{\mathrm{m.b.}}\right\rangle. \tag{FDT}
$$

In the next two subsections, we apply this general formula to simple observables in order to derive various stationary fluctuation-dissipation relations. Incidentally, the discrete version of the Fokker-Planck equation can be derived through the Kramers-Moyal expansion, considering the more general nonstationary version of the above equation and performing the Taylor expansion in $\eta$ and repeated integrations by parts (Van Kampen, 1992; Gardiner, 2009; Risken, 1984; Radons et al., 1990; Leen & Moody, 1993).

## 2.1 FIRST FLUCTUATION-DISSIPATION RELATION

Applying the master equation (FDT) to the linear observable,

$$
\langle \boldsymbol{\theta}\rangle = \left\langle \left[\!\left[\boldsymbol{\theta} - \eta\boldsymbol{\nabla}f^{\mathcal{B}}\left(\boldsymbol{\theta}\right)\right]\!\right]_{\mathrm{m.b.}}\right\rangle = \langle \boldsymbol{\theta}\rangle - \eta\langle\boldsymbol{\nabla}f\left(\boldsymbol{\theta}\right)\rangle. \tag{7}
$$

We thus have

$$
\langle \boldsymbol{\nabla}f\rangle = 0. \tag{8}
$$

This is natural because there is no particular direction that the gradient picks on average as the model parameter stochastically bounces around the local minimum or, more generally, wanders around the loss-function landscape according to the stationary distribution.

Performing similar algebra for the quadratic observable $\langle\theta_i\theta_j\rangle$ yields

$$
\langle\theta_i\left(\partial_j f\right)\rangle + \langle\left(\partial_i f\right)\theta_j\rangle = \eta\left\langle\widetilde{C}_{i,j}\right\rangle. \tag{9}
$$

In particular, taking the trace of this matrix-form relation, we obtain

$$
\langle\boldsymbol{\theta}\cdot\left(\boldsymbol{\nabla}f\right)\rangle = \frac{1}{2}\eta\left\langle\mathrm{Tr}\,\widetilde{C}\right\rangle. \tag{FDR1}
$$

More generally, in the case of SGD with momentum $\mu$ and dampening $\nu$, whose update equation is given by

$$
\begin{aligned}
\mathbf{v}(t+1) &= \mu\mathbf{v}(t) - (1-\nu)\boldsymbol{\nabla}f^{\mathcal{B}}\left[\boldsymbol{\theta}(t)\right], \tag{10} \\
\boldsymbol{\theta}(t+1) &= \boldsymbol{\theta}(t) + \eta\mathbf{v}(t+1), \tag{11}
\end{aligned}
$$

a similar derivation yields (see Appendix A)

$$
\langle\boldsymbol{\theta}\cdot\left(\boldsymbol{\nabla}f\right)\rangle = \frac{(1+\mu)}{2(1-\nu)}\eta\left\langle\mathbf{v}^2\right\rangle. \tag{FDR1'}
$$

The last equation reduces to the equation (FDR1) when $\mu = \nu = 0$ with $\mathbf{v} = -\boldsymbol{\nabla}f^{\mathcal{B}}$. Also note that $\langle\boldsymbol{\theta}\cdot\left(\boldsymbol{\nabla}f\right)\rangle = \langle\left(\boldsymbol{\theta} - \boldsymbol{\theta}_{\mathrm{c}}\right)\cdot\left(\boldsymbol{\nabla}f\right)\rangle$ for an arbitrary constant vector $\boldsymbol{\theta}_{\mathrm{c}}$ because of the equation (8).

This first fluctuation-dissipation relation is easy to evaluate on the fly during training, exactly holds without any approximation if sampled well from the stationary distribution, and can thus be used as the standard metric to check if learning has plateaued, just as similar relations can be used to check equilibration in Monte Carlo simulations of physical systems (Santen & Krauth, 2000). [It should be cautioned, however, that the fluctuation-dissipation relations are necessary but not sufficient to ensure stationarity (Odriozola & Berthier, 2011).] Such a metric can in turn be used to schedule changes in hyperparameters, as shall be demonstrated in Section 3.3.

## 2.2 SECOND FLUCTUATION-DISSIPATION RELATION

Applying the master equation (FDT) on the full-batch loss function and Taylor-expanding it in the learning rate $\eta$ yields the closed-form expression

$$
\begin{aligned}
\langle f\left(\boldsymbol{\theta}\right)\rangle &= \left\langle \left[\!\left[ f\left[\boldsymbol{\theta} - \eta \boldsymbol{\nabla} f^{\mathcal{B}}\left(\boldsymbol{\theta}\right)\right]\right]\!\right]_{\mathrm{m.b.}} \right\rangle \\
&= \left\langle f + \sum_{k=1}^{\infty} \frac{(-\eta)^k}{k!} \sum_{i_1,i_2,\ldots,i_k=1}^{P} \left(\partial_{i_1}\partial_{i_2}\cdots\partial_{i_k} f\right) \left[\!\left[\left(\partial_{i_1} f^{\mathcal{B}}\right)\left(\partial_{i_2} f^{\mathcal{B}}\right)\cdots\left(\partial_{i_k} f^{\mathcal{B}}\right)\right]\!\right]_{\mathrm{m.b.}} \right\rangle \\
&= \langle f \rangle - \eta \left\langle (\boldsymbol{\nabla} f)^2 \right\rangle + \sum_{k=2}^{\infty} \frac{(-\eta)^k}{k!} \left\langle \sum_{i_1,i_2,\ldots,i_k=1}^{P} F_{i_1,i_2,\ldots,i_k} \widetilde{C}_{i_1,i_2,\ldots,i_k} \right\rangle
\end{aligned}
\tag{12}
$$

where we recalled the equation (4) and introduced

$$
F_{i_1,i_2,\ldots,i_k}\left(\boldsymbol{\theta}\right) \equiv \partial_{i_1}\partial_{i_2}\cdots\partial_{i_k} f\left(\boldsymbol{\theta}\right) .
\tag{13}
$$

In particular, $H_{i,j}\left(\boldsymbol{\theta}\right) \equiv F_{i,j}\left(\boldsymbol{\theta}\right)$ is the Hessian matrix. Reorganizing terms, we obtain

$$
\left\langle (\boldsymbol{\nabla} f)^2 \right\rangle = \frac{\eta}{2} \left\langle \mathrm{Tr}\left(\boldsymbol{H}\widetilde{C}\right) \right\rangle - \eta^2 \left[ \sum_{k=3}^{\infty} \frac{(-\eta)^{k-3}}{k!} \left\langle \sum_{i_1,i_2,\ldots,i_k=1}^{P} F_{i_1,i_2,\ldots,i_k} \widetilde{C}_{i_1,i_2,\ldots,i_k} \right\rangle \right] .
\tag{FDR2}
$$

In the case of SGD with momentum and dampening, the left-hand side is replaced by $(1 - \nu)\left\langle (\boldsymbol{\nabla} f)^2 \right\rangle - \mu \langle \mathbf{v}\cdot\boldsymbol{\nabla} f\rangle$ and $\widetilde{C}_{i_1,i_2,\ldots,i_k}$ by more hideous expressions (see Appendix A).

We can extract at least two types of information on the loss-function landscape by evaluating the dependence of the left-hand side, $G(\eta) \equiv \left\langle (\boldsymbol{\nabla} f)^2 \right\rangle$, on the learning rate $\eta$. First, in the small learning rate regime, the value of $2G(\eta)/\eta$ approximates $\mathrm{Tr}\left(\boldsymbol{H}\widetilde{C}\right)$ around a local ravine. Second, nonlinearity of $G(\eta)$ at higher $\eta$ indicates discernible effects of anharmonicity. In such a regime, the Hessian matrix $\boldsymbol{H}$ cannot be approximated as constant (which also implies that $\{F_{i_1,i_2,\ldots,i_k}\}_{k>2}$ are nontrivial) and/or the noise two-point matrix $\widetilde{C}$ cannot be regarded as constant. Such nonlinearity especially indicates the breakdown of the harmonic approximation, that is, the quadratic truncation of the loss-function landscape, often used to analyze the regime explored at small learning rates.

## 2.3 REMARKS

### 2.3.1 INTUITION WITHIN THE HARMONIC APPROXIMATION

In order to gain some intuition about the fluctuation-dissipation relations, let us momentarily employ the harmonic approximation, i.e., assume that there is a local minimum of the loss function at $\boldsymbol{\theta} = \boldsymbol{\theta}^\star$ and retain only up to quadratic terms of the Taylor expansions around it: $f(\boldsymbol{\theta}) \approx f_0 + \frac{1}{2}\sum_{i,j=1}^{P} h_{i,j}(\theta_i - \theta_i^\star)(\theta_j - \theta_j^\star)$. Within this approximation, $\langle \boldsymbol{\theta}\cdot(\boldsymbol{\nabla} f)\rangle = \langle(\boldsymbol{\theta} - \boldsymbol{\theta}^\star)\cdot(\boldsymbol{\nabla} f)\rangle \approx 2\langle f - f_0\rangle$. The relation (FDR1) then becomes $\langle f - f_0\rangle \approx \frac{1}{4}\eta\left\langle \mathrm{Tr}\widetilde{C}\right\rangle$, linking the height of the noise ball to the noise amplitude. This is in line with, for instance, the theorem 4.6 of the reference Bottou et al. (2018) and substantiates the analogy between SGD and simulated annealing, with the learning rate $\eta$ – multiplied by $\mathrm{Tr}\widetilde{C}$ – playing the role of temperature (Bottou, 1991).

### 2.3.2 HIGHER-ORDER RELATIONS

Additional relations can be derived by repeating similar calculations for higher-order observables. For example, at the cubic order,

$$
\langle \theta_i\theta_j\left(\partial_k f\right) + \theta_i\left(\partial_j f\right)\theta_k + \left(\partial_i f\right)\theta_j\theta_k\rangle = \eta\left\langle \theta_i\widetilde{C}_{j,k} + \theta_j\widetilde{C}_{k,i} + \theta_k\widetilde{C}_{i,j}\right\rangle - \eta^2\left\langle \widetilde{C}_{i,j,k}\right\rangle .
\tag{14}
$$

The systematic investigation of higher-order relations is relegated to future work.

### 2.3.3 SGD≠SDE

There is no limit in which SGD asymptotically reduces to the stochastic differential equation (SDE). In order to take such a limit with continuous time differential $\mathrm{d}t \to 0^+$, each SGD update must become infinitesimal. One may thus try $\mathrm{d}t \equiv \eta \to 0^+$, as in recent work adapting the view that SGD=SDE (Mandt et al., 2015; Li et al., 2015; Mandt et al., 2017; Li et al., 2017; Smith & Le, 2018; Chaudhari & Soatto, 2017; Jastrzebski et al., 2017; Zhu et al., 2018; An et al., 2018). But this in turn forces the noise vector with zero mean, $\boldsymbol{\nabla} f^{\mathcal{B}} - \boldsymbol{\nabla} f$, to be multiplied by $\mathrm{d}t$. This is in contrast to the scaling $\sqrt{\mathrm{d}t}$ needed for the standard machinery of SDE – Itô-Stratonovich calculus and all that – to apply; the additional factor of $\mathrm{d}t^{1/2}$ makes the effective noise covariance be suppressed by $\mathrm{d}t$ and the resulting equation in the continuous-time limit, if anything, would just be an ordinary differential equation without noise[2] [unless noise with the proper scaling is explicitly added as in stochastic gradient Langevin dynamics (Welling & Teh, 2011; Teh et al., 2016) and natural Langevin dynamics (Marceau-Caron & Ollivier, 2017; Nado et al., 2018)].

In short, the recent work views $\eta = \sqrt{\eta}\sqrt{\mathrm{d}t}$ and sends $\mathrm{d}t \to 0^+$ while pretending that $\eta$ is finite, which is inconsistent. This is not just a technical subtlety. When unjustifiably passing onto the continuous-time Fokker-Planck equation, the diffusive term is incorrectly governed by the connected two-point noise matrix $C_{i,j}(\boldsymbol{\theta}) \equiv \widetilde{C}_{i,j}(\boldsymbol{\theta}) - [\partial_i f(\boldsymbol{\theta})][\partial_j f(\boldsymbol{\theta})]$ rather than the full two-point noise matrix $\widetilde{C}_{i,j}(\boldsymbol{\theta})$ that appears herein.[3] We must instead employ the discrete-time version of the Fokker-Planck equation derived in references Van Kampen (1992); Gardiner (2009); Risken (1984); Radons et al. (1990); Leen & Moody (1993), as has been followed in the equation (6).

### 2.3.4 ON STATIONARITY

In contrast to statistical mechanics where an equilibrium state is dictated by a handful of thermodynamic variables, in machine learning a stationary state generically depends not only on hyperparameters but also on a part of its learning history. The stationarity assumption made herein, which is codified in the equation (6), is weaker than the typicality assumption underlying statistical mechanics and can hold even in the presence of lingering memory. In the full-batch limit $|\mathcal{B}| = N_\mathrm{s}$, for instance, any distribution delta-peaked at a local minimum is stationary. For sufficiently small learning rates $\eta$ as well, it is natural to expect multiple stationary distributions that form disconnected ponds around these minima, which merge upon increasing $\eta$ and fragment upon decreasing $\eta$.

It is beyond the scope of the present paper to formulate conditions under which stationary distributions exist. Indeed, if the formulation were too generic, there could be counterexamples to such a putative existence statement. A case in point is a model with the unregularized cross entropy loss, whose model parameters keep cascading toward infinity in order to sharpen its softmax output (Neyshabur et al., 2014; 2017) with logarithmically diverging $\boldsymbol{\theta}^2$ (Soudry et al., 2018). It would be interesting to see if there are any other nontrivial caveats.

## 3 EMPIRICAL TESTS

In this section we empirically bear out our theoretical claims in the last section. To this end, two simple models of supervised learning are used (see Appendix B for full specifications): a multilayer perceptron (MLP) learning patterns in the MNIST training data (LeCun et al., 1998) through SGD without momentum and a convolutional neural network (CNN) learning patterns in the CIFAR-10 training data (Krizhevsky & Hinton, 2009) through SGD with momentum $\mu = 0.9$. For both models, the mini-batch size is set to be $|\mathcal{B}| = 100$, and the training data are shuffled at each epoch $t = \frac{N_\mathrm{s}}{|\mathcal{B}|}\hat{t}_\mathrm{epoch}$ with $\hat{t}_\mathrm{epoch} \in \mathbb{N}$. In order to avoid the overfitting cascade mentioned in Section 2.3.4, the $L^2$-regularization term $\frac{1}{2}\lambda\boldsymbol{\theta}^2$ with the weight decay $\lambda = 0.01$ is included in the loss function $f$.

---

[2] One may try to evade this by employing the $1/|\mathcal{B}|$-scaling of the connected noise covariant matrix, but that would then enforces $|\mathcal{B}| \to 0^+$ as $\mathrm{d}t \to 0^+$, which is unphysical.

[3] Heuristically, $(\boldsymbol{\nabla} f)^2 \sim \eta \boldsymbol{H}\widetilde{\boldsymbol{C}}$ for small $\eta$ due to the relation FDR2, and one may thus neglect the difference between $\widetilde{\boldsymbol{C}}$ and $\boldsymbol{C}$, and hence justify the naive use of SDE, when $\eta\boldsymbol{H} \ll 1$ and the Gaussian-noise assumption holds. In the similar vein, the reference Li et al. (2015) proves faster convergence between SGD and SDE when the term proportional to $\eta\boldsymbol{\nabla}(\boldsymbol{\nabla} f)^2$ is added to the gradient.

Before proceeding further, let us define the half-running average of an observable $\mathcal{O}$ as

$$\overline{\mathcal{O}}(t) \equiv \frac{1}{t - t_0} \sum_{t'=t_0+1}^{t} \mathcal{O}(t') \quad \text{with} \quad t_0 = \lfloor t/2 \rfloor . \tag{15}$$

This is the average of the observable up to the time step $t$, with the initial half discarded as containing transient. If SGD drives the distribution of the model parameters to stationarity at long time, then

$$\lim_{t \to \infty} \overline{\mathcal{O}}(t) = \langle \mathcal{O} \rangle . \tag{16}$$

### 3.1 FIRST FLUCTUATION-DISSIPATION RELATION AND EQUILIBRATION

In order to assess the proximity to stationarity, define

$$\mathcal{O}_{\mathrm{L}} \equiv \boldsymbol{\theta} \cdot \left( \boldsymbol{\nabla} f^{\mathcal{B}} \right) \quad \text{and} \quad \mathcal{O}_{\mathrm{R}} \equiv \frac{(1+\mu)}{2(1-\nu)} \eta \mathbf{v}^2 \tag{17}$$

(with $\mathbf{v}$ replaced by $-\boldsymbol{\nabla} f^{\mathcal{B}}$ for SGD without momentum).[4] Both of these observables can easily be measured on the fly at each time step during training and, according to the relation (FDR1'), the running averages of these two observables should converge to each other upon equilibration.

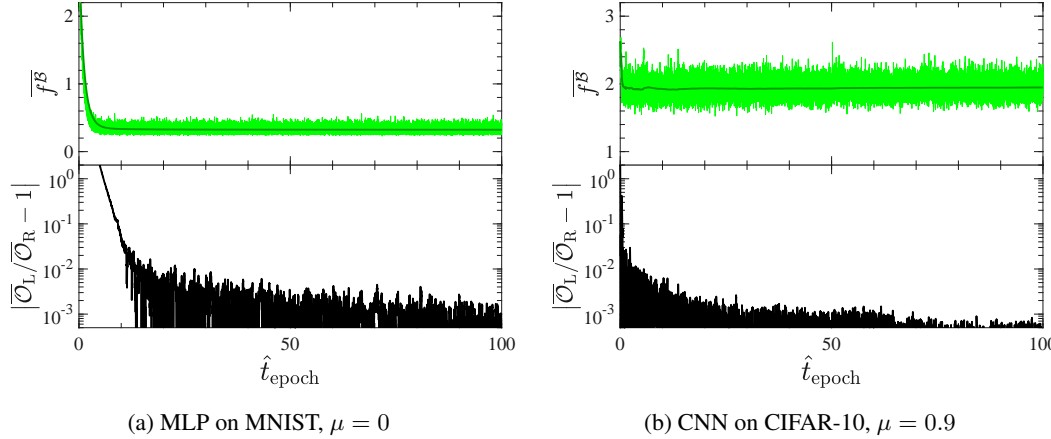

(a) MLP on MNIST, $\mu = 0$          (b) CNN on CIFAR-10, $\mu = 0.9$

Figure 1: Approaches toward stationarity during the initial trainings for the MLP on the MNIST data (a) and for the CNN on the CIFAR-10 data (b). Top panels depict the half-running average $\overline{f^{\mathcal{B}}}(t)$ (dark green) and the instantaneous value $f^{\mathcal{B}}(t)$ (light green) of the mini-batch loss. Bottom panels depict the convergence of the half-running averages of the observables $\mathcal{O}_{\mathrm{L}} = \boldsymbol{\theta} \cdot \boldsymbol{\nabla} f^{\mathcal{B}}$ and $\mathcal{O}_{\mathrm{R}} = \frac{(1+\mu)}{2(1-\nu)} \eta \mathbf{v}^2$, whose stationary-state averages should agree according to the relation (FDR1').

In order to verify this claim, we first train the model with the learning rate $\eta = 0.1$ for $\hat{t}_{\mathrm{epoch}}^{\mathrm{total}} = 100$ epochs, that is, for $t^{\mathrm{total}} = \frac{N_{\mathrm{s}}}{|\mathcal{B}|} \hat{t}_{\mathrm{epoch}}^{\mathrm{total}} = 100 \frac{N_{\mathrm{s}}}{|\mathcal{B}|}$ time steps. As shown in the figure 1, the observables $\overline{\mathcal{O}}_{\mathrm{L}}(t)$ and $\overline{\mathcal{O}}_{\mathrm{R}}(t)$ converge to each other. We then take the model at the end of the initial 100-epoch training and sequentially train it further at various learning rates $\eta$ (see Appendix B). The observables $\overline{\mathcal{O}}_{\mathrm{L}}(t)$ and $\overline{\mathcal{O}}_{\mathrm{R}}(t)$ again converge to each other, as plotted in the figure 2. Note that the smaller the learning rate is, the longer it takes to equilibrate.

### 3.2 SECOND FLUCTUATION-DISSIPATION RELATION AND SHAPE OF LOSS-FUNCTION LANDSCAPE

In order to assess the loss-function landscape information from the relation (FDR2), define

$$\mathcal{O}_{\mathrm{FB}} \equiv (1-\nu) \left( \boldsymbol{\nabla} f \right)^2 - \mu \mathbf{v} \cdot \boldsymbol{\nabla} f^{\mathcal{B}} \tag{18}$$

---

[4]If the model parameter $\boldsymbol{\theta}$ happens to fluctuate around large values, for numerical accuracy, one may want to replace $\mathcal{O}_{\mathrm{L}} = \boldsymbol{\theta} \cdot \left( \boldsymbol{\nabla} f^{\mathcal{B}} \right)$ by $\left( \boldsymbol{\theta} - \boldsymbol{\theta}_{\mathrm{c}} \right) \cdot \left( \boldsymbol{\nabla} f^{\mathcal{B}} \right)$ where a constant vector $\boldsymbol{\theta}_{\mathrm{c}}$ approximates the vector around which $\boldsymbol{\theta}$ fluctuates at long time.

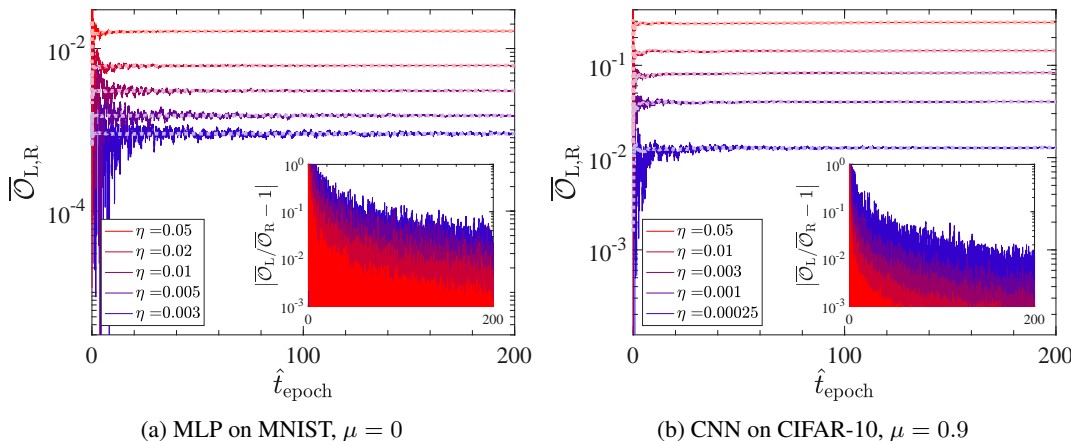

(a) MLP on MNIST, $\mu = 0$      (b) CNN on CIFAR-10, $\mu = 0.9$

Figure 2: Approaches toward stationarity during the sequential runs for various learning rates $\eta$, seen through the half-running averages of the observables $\mathcal{O}_{\mathrm{L}} = \boldsymbol{\theta} \cdot \boldsymbol{\nabla} f^{\mathcal{B}}$ (solid) and $\mathcal{O}_{\mathrm{R}} = \frac{(1+\mu)}{2(1-\nu)} \eta \mathbf{v}^2$ (dotted light-colored). They agree at sufficiently long times but the relaxation time to reach such a stationary regime increases as the learning rate $\eta$ decreases.

(with the second term nonexistent for SGD without momentum).[5] Note that $(\boldsymbol{\nabla} f)^2$ is a full-batch – not mini-batch – quantity. Given its computational cost, here we measure this first term only at the end of each epoch and take the half-running average over these sparse sample points, discarding the initial half of the run.

The half-running average of the full-batch observable $\overline{\mathcal{O}}_{\mathrm{FB}}$ at the end of sufficiently long training, which is a good proxy for $\langle \mathcal{O}_{\mathrm{FB}} \rangle$, is plotted in the figure 3 as a function of the learning rate $\eta$. As predicted by the relation (FDR2), at small learning rates $\eta$, the observable $\langle \mathcal{O}_{\mathrm{FB}} \rangle$ approaches zero; its slope – divided by $\left\langle \mathrm{Tr}\, \widetilde{C} \right\rangle$ if preferred – measures the magnitude of the Hessian matrix, component-wise averaged over directions in which the noise preferentially fluctuates. Meanwhile, nonlinearity at higher learning rates $\eta$ measures the degree of anharmonicity experienced over the distribution $p_{\mathrm{ss}}(\boldsymbol{\theta})$. We see that anharmonic effects are pronounced especially for the CNN on the CIFAR-10 data even at moderately small learning rates. This invalidates the use of the quadratic harmonic approximation for the loss-function landscape and/or the assumption of the constant noise matrix for this model except at very small learning rates.

### 3.3 First fluctuation-dissipation relation and learning-rate schedules

Saturation of the relation (FDR1) suggests the learning stationarity, at which point it might be wise to decrease the learning rate $\eta$. Such scheduling is often carried out in an ad hoc manner but we can now algorithmize this procedure as follows:

1. Evaluate the half-running averages $\overline{\mathcal{O}}_{\mathrm{L}}(t)$ and $\overline{\mathcal{O}}_{\mathrm{R}}(t)$ at the end of each epoch.

2. If $\left| \dfrac{\overline{\mathcal{O}}_{\mathrm{L}}(t)}{\overline{\mathcal{O}}_{\mathrm{R}}(t)} - 1 \right| < X$, then decrease the learning rate as $\eta \to (1 - Y)\eta$ and also set $t = 0$ for the purpose of evaluating half-running averages.

Here, two scheduling hyperparameters $X$ and $Y$ are introduced, which control the threshold for saturation of the relation (FDR1) and the amount of decrease in the learning rate, respectively.

Plotted in the figure 4 are results for SGD without momentum, with the Xavier initialization (Glorot & Bengio, 2010) and training through (i) preset training schedule with decrease of the learning rate by a factor of 10 for each 100 epochs, (ii) an adaptive scheduler with $X = 0.01$ (1% threshold) and

---

[5]For the second term, in order to ensure that $\lim_{t \to \infty} \overline{\mathbf{v} \cdot \boldsymbol{\nabla} f^{\mathcal{B}}}(t) = \lim_{t \to \infty} \overline{\mathbf{v} \cdot \boldsymbol{\nabla} f}(t)$, we measure the half-running average of $\mathbf{v}(t) \cdot \boldsymbol{\nabla} f^{\mathcal{B}}[\boldsymbol{\theta}(t)]$ and not $\mathbf{v}(t+1) \cdot \boldsymbol{\nabla} f^{\mathcal{B}}[\boldsymbol{\theta}(t)]$.

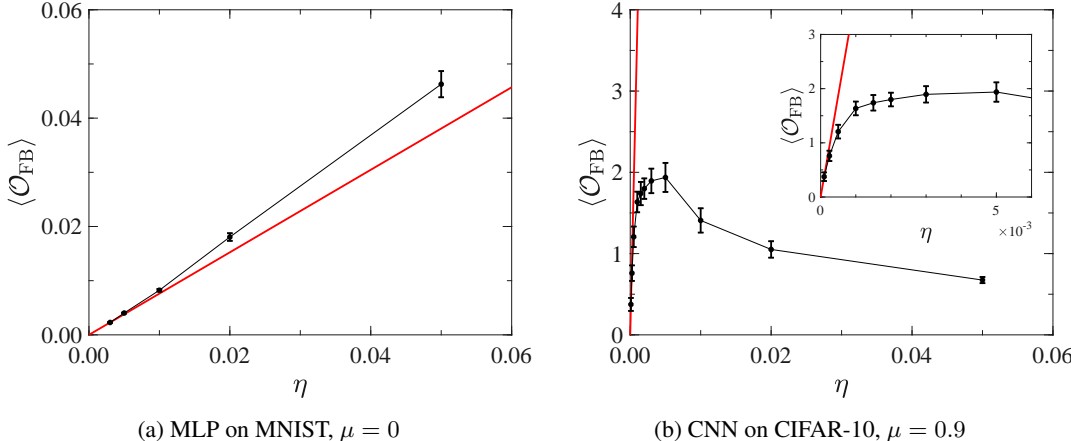

(a) MLP on MNIST, $\mu = 0$        (b) CNN on CIFAR-10, $\mu = 0.9$

Figure 3: The stationary-state average of the full-batch observable $\mathcal{O}_{\mathrm{FB}}$ as a function of the learning rate $\eta$, estimated through half-running averages. Dots and error bars denote mean values and $95\%$ confidence intervals over several distinct runs, respectively. The straight red line connects the origin and the point with the smallest $\eta$ explored. (a) For the MLP on the MNIST data, linear dependence on $\eta$ for $\eta \lesssim 0.01$ supports the validity of the harmonic approximation there. (b) For the CNN on the CIFAR-10 data, anharmonicity is pronounced even down to $\eta \sim 0.001$.

$Y = 0.1$ ($10\%$ decrease), and (iii) the AMSGrad algorithm (J. Reddi et al., 2018) with the default hyperparameters. The adaptive scheduler attains comparable accuracies with the preset scheduling at long time and outperforms AMSGrad (see Appendix C for additional simulations).

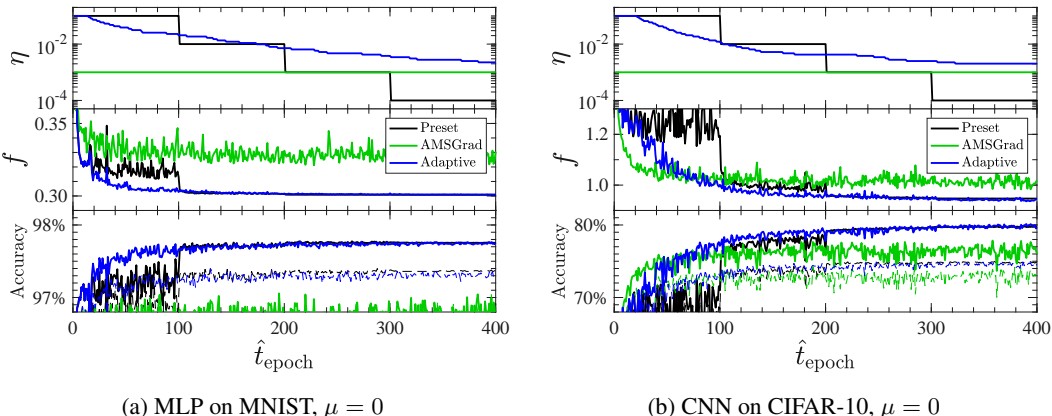

(a) MLP on MNIST, $\mu = 0$        (b) CNN on CIFAR-10, $\mu = 0$

Figure 4: Comparison of preset training schedule (black) and adaptive training schedule (blue), employing SGD without momentum both for the MLP on the MNIST data (a) and the CNN on the CIFAR-10 data (b), along with the AMSGrad algorithm (green). From top to bottom, plotted are the learning rate $\eta$, the full-batch training loss $f$, and prediction accuracies on the training-set images (solid) and the 10000 test-set images (dashed).

These two scheduling methods span different subspaces of all the possible schedules. The adaptive scheduling method proposed herein has a theoretical grounding and in practice much less dimensionality for tuning of scheduling hyperparameters than the presetting method, thus ameliorating the optimization of scheduling hyperparameters. The systematic comparison between the two scheduling methods for state-of-the-arts architectures, and also the comparison with the AMSGrad algorithm for natural language processing tasks, could be a worthwhile avenue to pursue in the future.

## 4 CONCLUSION

In this paper, we have derived the fluctuation-dissipation relations with no assumptions other than stationarity of the probability distribution. These relations hold exactly even when the noise is non-Gaussian and the loss function is nonconvex. The relations have been empirically verified and used to probe the properties of the loss-function landscapes for the simple models. The relations further have resulted in the algorithm to adaptively set learning-rate schedule on the fly rather than presetting it in an ad hoc manner. In addition to systematically testing the performance of this adaptive scheduling algorithm, it would be interesting to investigate non-Gaussianity and noncovexity in more details through higher-point observables, both analytically and numerically. It would also be interesting to further elucidate the physics of machine learning by extending our formalism to incorporate nonstationary dynamics, linearly away from stationarity (Onsager, 1931; Green, 1954; Kubo, 1957) and beyond (Jarzynski, 1997; Crooks, 1999), so that it can in particular properly treat overfitting cascading dynamics and time-dependent sample distributions.

### ACKNOWLEDGMENTS

The author thanks Ludovic Berthier, Léon Bottou, Guy Gur-Ari, Kunihiko Kaneko, Ari Morcos, Dheevatsa Mudigere, Yann Ollivier, Yuandong Tian, and Mark Tygert for discussions. Special thanks go to Daniel Adam Roberts who prompted the practical application of the fluctuation-dissipation relations, leading to the adaptive method in Section 3.3.

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

## A  SGD WITH MOMENTUM AND DAMPENING

For SGD with momentum $\mu$ and dampening $\nu$, the update equation is given by

$$
\begin{aligned}
\mathbf{v}(t+1) &= \mu \mathbf{v}(t) - (1-\nu)\boldsymbol{\nabla} f^{\mathcal{B}}\left[\boldsymbol{\theta}(t)\right], & (19) \\
\boldsymbol{\theta}(t+1) &= \boldsymbol{\theta}(t) + \eta \mathbf{v}(t+1). & (20)
\end{aligned}
$$

Here $\mathbf{v} = \{v_i\}_{i=1,\dots,P}$ is the velocity and $\eta > 0$ the learning rate; SGD without momentum is the special case with $\mu = 0$. Again hypothesizing the existence of a stationary-state distribution $p_{\mathrm{ss}}(\boldsymbol{\theta}, \mathbf{v})$, the stationary-state average of an observable $\mathcal{O}(\boldsymbol{\theta}, \mathbf{v})$ is defined as

$$
\langle \mathcal{O}(\boldsymbol{\theta}, \mathbf{v}) \rangle \equiv \int \mathrm{d}\boldsymbol{\theta}\mathrm{d}\mathbf{v}\, p_{\mathrm{ss}}(\boldsymbol{\theta}, \mathbf{v})\, \mathcal{O}(\boldsymbol{\theta}, \mathbf{v}). \tag{21}
$$

Just as in the main text, from the assumed stationarity follows the master equation for SGD with momentum and dampening

$$
\langle \mathcal{O}(\boldsymbol{\theta}, \mathbf{v}) \rangle = \Big\langle \llbracket \mathcal{O}\left\{\boldsymbol{\theta} + \eta\left[\mu\mathbf{v} - (1-\nu)\boldsymbol{\nabla} f^{\mathcal{B}}(\boldsymbol{\theta})\right], \ \mu\mathbf{v} - (1-\nu)\boldsymbol{\nabla} f^{\mathcal{B}}(\boldsymbol{\theta})\right\}\rrbracket_{\mathrm{m.b.}} \Big\rangle. \tag{22}
$$

For the linear observables,

$$
\langle \mathbf{v} \rangle = \mu \langle \mathbf{v} \rangle - (1-\nu)\langle \boldsymbol{\nabla} f(\boldsymbol{\theta}) \rangle \tag{23}
$$

and

$$
\langle \boldsymbol{\theta} \rangle = \langle \boldsymbol{\theta} \rangle + \eta\left[\mu \langle \mathbf{v} \rangle - (1-\nu)\langle \boldsymbol{\nabla} f(\boldsymbol{\theta}) \rangle\right] = \langle \boldsymbol{\theta} \rangle + \eta \langle \mathbf{v} \rangle, \tag{24}
$$

thus

$$
\langle \mathbf{v} \rangle = 0 \quad \text{and} \quad \langle \boldsymbol{\nabla} f \rangle = 0. \tag{25}
$$

For the quadratic observables

$$
\langle v_i v_j \rangle = \mu^2 \langle v_i v_j \rangle + (1-\nu)^2 \left\langle \widetilde{C}_{i,j} \right\rangle - (1-\nu)\mu\left[\langle v_i (\partial_j f) \rangle + \langle (\partial_i f) v_j \rangle\right], \tag{26}
$$

$$
\langle v_i \theta_j \rangle - \eta \langle v_i v_j \rangle = \mu \langle v_i \theta_j \rangle - (1-\nu)\langle (\partial_i f)\theta_j \rangle, \tag{27}
$$

and

$$
(1-\nu)\left[\langle \theta_i (\partial_j f) \rangle + \langle (\partial_i f)\theta_j \rangle\right] - \mu\left(\langle \theta_i v_j \rangle + \langle v_i \theta_j \rangle\right) = \eta \langle v_i v_j \rangle. \tag{28}
$$

Note that the relations (26) and (27) are trivially satisfied at each time step if the left-hand side observables are evaluated at one step ahead and thus their being satisfied for running averages has nothing to do with equilibration [the same can be said about the relation (23)]; the only nontrivial relation is the equation (28), which is a consequence of setting $\langle \theta_i \theta_j \rangle$ constant of time. After taking traces and some rearrangement, we obtain the relation (FDR1') in the main text.

For the full-batch loss function, the algebra similar to the one in the main text yields

$$
\left[(1-\nu)\left\langle (\boldsymbol{\nabla} f)^2 \right\rangle - \mu \langle \mathbf{v} \cdot \boldsymbol{\nabla} f \rangle\right] \tag{29}
$$

$$
= \eta \left\langle \sum_{i,j=1}^{P} H_{i,j}\left\{(1-\nu)^2 \widetilde{C}_{i,j} - \mu(1-\nu)\left[v_i (\partial_j f) + (\partial_i f) v_j\right] + \mu^2 v_i v_j\right\} \right\rangle + O(\eta^2).
$$

## B MODELS AND SIMULATION PROTOCOLS

### B.1 MLP ON MNIST THROUGH SGD WITHOUT MOMENTUM

The MNIST training data consist of $N_\mathrm{s} = 60000$ black-white images of hand-written digits with 28-by-28 pixels (LeCun et al., 1998). We preprocess the data through an affine transformation such that their mean and variance (over both the training data and pixels) are zero and one, respectively.

Our multilayer perceptron (MLP) consists of a 784-dimensional input layer followed by a hidden layer of 200 neurons with ReLU activations, another hidden layer of 200 neurons with ReLU activations, and a 10-dimensional output layer with the softmax activation. The model performance is evaluated by the cross-entropy loss supplemented by the $L^2$-regularization term $\frac{1}{2}\lambda\boldsymbol{\theta}^2$ with the weight decay $\lambda = 0.01$.

Throughout the paper, the MLP is trained on the MNIST data through SGD without momentum. The data are shuffled at each epoch with the mini-batch size $|\mathcal{B}| = 100$.

The MLP is initialized through the Xavier method (Glorot & Bengio, 2010) and trained for $\hat{t}_\mathrm{epoch}^\mathrm{total} = 100$ epochs with the learning rate $\eta = 0.1$. We then sequentially train it with $(\eta, \hat{t}_\mathrm{epoch}^\mathrm{total}) = (0.05, 500) \rightarrow (0.02, 500) \rightarrow (0.01, 500) \rightarrow (0.005, 1000) \rightarrow (0.003, 1000)$. This sequential-run protocol is carried out with 4 distinct seeds for the random-number generator used in data shuffling, all starting from the common model parameter attained at the end of the initial 100-epoch run. The figure 2 depicts trajectories for one particular seed, while the figure 3 plots means and error bars over these distinct seeds.

### B.2 CNN ON CIFAR-10 THROUGH SGD WITH MOMENTUM

The CIFAR-10 training data consist of $N_\mathrm{s} = 50000$ color images of objects – divided into ten categories – with 32-by-32 pixels in each of 3 color channels, each pixel ranging in $[0, 1]$ (Krizhevsky & Hinton, 2009). We preprocess the data through uniformly subtracting 0.5 and multiplying by 2 so that each pixel ranges in $[-1, 1]$.

In order to describe the architecture of our convolutional neural network (CNN) in detail, let us associate a tuple $[F, C, S, P; M]$ to a convolutional layer with filter width $F$, a number of channels $C$, stride $S$, and padding $P$, followed by ReLU activations and a max-pooling layer of width $M$. Then, as in the demo at Karpathy (2014), our CNN consists of a $(32, 32, 3)$ input layer followed by a convolutional layer with $[5, 16, 1, 2; 2]$, another convolutional layer with $[5, 20, 1, 2; 2]$, yet another convolutional layer with $[5, 20, 1, 2; 2]$, and finally a fully-connected 10-dimensional output layer with the softmax activation. The model performance is evaluated by the cross-entropy loss supplemented by the $L^2$-regularization term $\frac{1}{2}\lambda\boldsymbol{\theta}^2$ with the weight decay $\lambda = 0.01$.

Throughout the paper (except in Section 3.3 where the adaptive scheduling method is tested for SGD without momentum), the CNN is trained on the CIFAR-10 data through SGD with momentum $\mu = 0.9$ and dampening $\nu = 0$. The data are shuffled at each epoch with the mini-batch size $|\mathcal{B}| = 100$.

The CNN is initialized through the Xavier method (Glorot & Bengio, 2010) and trained for $\hat{t}_\mathrm{epoch}^\mathrm{total} = 100$ epochs with the learning rate $\eta = 0.1$. We then sequentially train it with $(\eta, \hat{t}_\mathrm{epoch}^\mathrm{total}) = (0.05, 200) \rightarrow (0.02, 200) \rightarrow (0.01, 200) \rightarrow (0.005, 400) \rightarrow (0.003, 400) \rightarrow (0.002, 400) \rightarrow (0.0015, 400) \rightarrow (0.001, 400) \rightarrow (0.0005, 800) \rightarrow (0.00025, 800) \rightarrow (0.0001, 800)$. At each junction of the sequence, the velocity $\mathbf{v}$ is zeroed. This sequential-run protocol is carried out with 16 distinct seeds for the random-number generator used in data shuffling, all starting from the common model parameter attained at the end of the initial 100-epoch run. The figure 2 depicts trajectories for one particular seed, while the figure 3 plots means and error bars over these distinct seeds.

## C  ADDITIONAL SIMULATIONS

### C.1  ADAM VERSUS AMSGRAD

Plotted in the figure S1 are the comparisons between Adam (Kingma & Ba, 2014) and AMS-Grad (J. Reddi et al., 2018) algorithms with the default hyperparameters $\alpha = 10^{-3}$, $(\beta_1, \beta_2) = (0.9, 0.999)$, and $\epsilon = 10^{-8}$. The AMSGrad algorithm marginally outperforms the Adam algorithm for the tasks at hand and thus the results with the AMSGrad are presented in the main text.

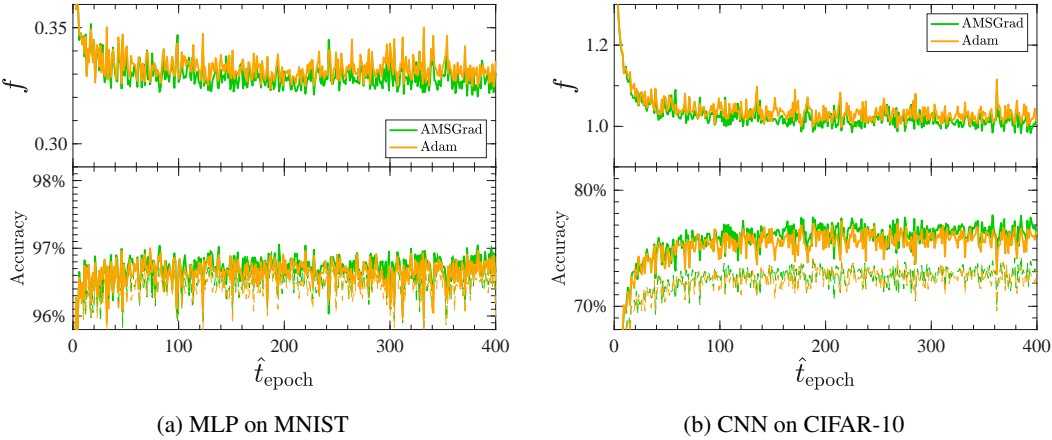

(a) MLP on MNIST                    (b) CNN on CIFAR-10

Figure S1: Comparison of AMSGrad (green) and Adam (orange) algorithms for the MLP on the MNIST data (a) and the CNN on the CIFAR-10 data (b). Top rows plot the full-batch training loss $f$ while bottom rows plot prediction accuracies on the training-set images (solid) and the 10000 test-set images (dashed).

### C.2  INITIAL ACCURACY GAIN WITH DIFFERENT SCHEDULING HYPERPARAMETERS

In the figure 4(a) for the MNIST classification task with the MLP, the proposed adaptive method with the scheduling hyperparameters $X = 0.01$ and $Y = 0.1$ outperforms the AMSGrad algorithm in terms of accuracy attained at long time and also exhibits a quick initial convergence. In the figure 4(b) for the CIFAR-10 classification task with the CNN, however, while the proposed adaptive method attains better accuracy at long time, its initial accuracy gain is visibly slower than the AMSGrad algorithm. This lag in initial accuracy gain can be ameliorated by choosing another combination of the scheduling hyperparameters, e.g., $X = 0.1$ and $Y = 0.3$, at the expense of degradation in generalization accuracy with respect to the original choice $X = 0.01$ and $Y = 0.1$. See the figure S2.

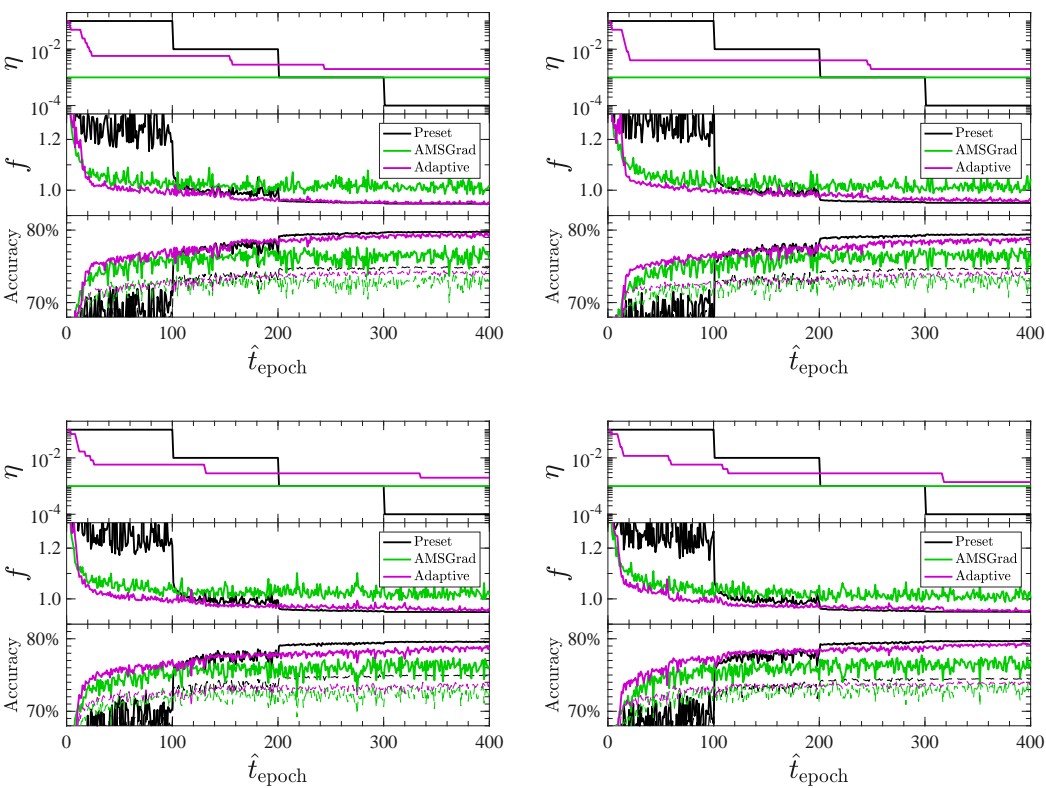

Figure S2: Comparison of preset training schedule (black) and adaptive training schedule (purple) – now with the scheduling hyperparameters $X = 0.1$ and $Y = 0.3$ – employing SGD without momentum, and the AMSGrad algorithm (green), for the CNN on the CIFAR-10 data with the same initial seed as in the main text (a) and three different initial seeds (b-d). From top to bottom, plotted are the learning rate $\eta$, the full-batch training loss $f$, and prediction accuracies on the training-set images (solid) and the 10000 test-set images (dashed).

