# OpenReview forum: "Fluctuation-dissipation relations for stochastic gradient descent"
_ICLR.cc/2019/Conference_

### Official Review · AnonReviewer3 · 2018-11-01
**A nice attempt to understand the stationary equilibrium of SGD but the paper not easy to follow**

**Rating:** 6
**Confidence:** 3

**Review:**

Understanding the stationary equilibrium helps to understand the practical performance of stochastic gradient descent. In this paper, the authors propose two fluctuation-dissipation relation to link measurable quantities and hyperparameters in the stochastic gradient descent algorithm. An advantage over the existing study is that the results here hold for any stationary state and do not need the analogy with continuous-time differential equations. Empirical results are also reported to verify these fluctuation relation.

Comments:

(1) I do not quite understand the second identity of (16). In particular, it seems that the authors replace the first two $\theta(t+1)$ with $\theta(t)$, and do not use this replacement for the third $\theta(t+1)$ (this was addressed by (1)).

(2) It would be helpful to the readers if the authors can give the deduction process of (12). It is not easy for me to understand how it holds.

(3) It is not clear to me how the second fluctuation-dissipation relation helps to determine the properties of loss function landscape.

(4) In Section 2.3.1, can you give some explanation for the harmonic approximation. Also the notation $\theta^*$ seems not to be defined.

---

> ### Author Response · Authors · 2018-11-14
> **Equation (6) made more transparent; equation (12) made more explicit; harmonic approximation defined.**
>
> Thank you very much for your clarifying comments! Please see below for our responses to your comments (and note our responses to other reviewers as well, especially the discussion with the Reviewer 1 which led to additional experiments).
>
>
> (1) We fully agree that the equation (6) was very confusing. Please see the revised version, wherein we have unrolled the derivation more explicitly. In particular, right before the equation (6), we now state how a probability distribution of model parameters evolves in general, which is used to make the second identity easier to follow; the last identity is just an integration over theta against the delta function. Please let us know if the derivation is still unclear.
>
>
> (2) We have added one middle step in the equation (12) so as to make the deduction process more transparent. We also explicitly allude to the facts that we are Taylor-expanding in the learning rate eta and that the equation (4) has been used. Please let us know if it is still not clear.
>
>
> (4) The first sentence of the Section 2.3.1 now explicitly states what we mean by the harmonic approximation and also defines theta^*.
>
>
> (3) The ways in which the second fluctuation-dissipation relation help to determine the properties of loss-function landscape have been stipulated right after the equation (FDR2) and in the second paragraph of Section 3.2 (along with the figure 3). Before explaining them below in a way slightly different from the draft, let us emphasize that properties exposed by (FDR2) are far from constituting the exhaustive list of loss-function landscape's properties which people would care [e.g., low eigenvalues of loss-function landscapes are essentially invisible to (FDR2)].
>
> (3-I) The Hessian H (~curvature) of a local minimum is one property of the loss-function landscape and Tr(H\tilde{C}) gives one particular slice of this information. Now, computing this quantity naively would entail the computational cost that scales as P^2 where P is a number of model parameters. The left-hand side of (FDR2), in contrast, has the computational cost that scales as P and hence allows the computation of Tr(H\tilde{C}) for large models that is more efficient than the direct computation.
>
> (3-II) The more nontrivial result is the fact that the nonlinearity of <grad(f)^2> as a function of the learning rate eta tells the breakdown of the idealized assumption on the loss-function landscape (such as the harmonic approximation, the breakdown of which reflects the degree of nonconvexity of the landscape). The inappropriateness (when so) of such idealized assumptions on the landscape is another property that (FDR2) exposes [which we found to be vividly pronounced for CIFAR-10 experiments even at fairly small learning rates: please see the figure 3(b)].
>
>
> We hope that all the confusing points have now been resolved and that, as a result, the paper is easier to follow than before. We thank you again and would appreciate further clarification if any.

---

### Official Review · AnonReviewer1 · 2018-11-02
**Inspired by statistical mechanics, the authors derive the stationary fluctuation-dissipation relations that link measurable quantities and hyperparameters in SGD. They further use the relations to set training schedule adaptively and analyze the loss-function landscape. However, the analysis about the stationarity assumption is insufficient and some experiments are weak.**

**Rating:** 5
**Confidence:** 4

**Review:**

The authors establish a stationary fluctuation-dissipation theorem and derive two specific fluctuation-dissipation relations. The authors use the first relation to check the stationarity and the second relation to delineate the shape of the loss-function landscape.
To verify their claim, the authors further use the relations to set the learning-rate schedule adaptively in SGD.

My major concerns are as follows.

1. The experiments in subsection 3.3 are not convincing. The authors compare the proposed adaptive training schedule with a preset training schedule. However, the improvement by the proposed schedule is insignificant.
To make this paper more convincing, the authors may want to compare the proposed adaptive training schedule with other approaches that have dynamic learning rates, such as those mentioned in [1].

2. The derived relations are based on the stationarity assumption. However, there are few discussions on when this assumption will hold. The authors may want to analyze the conditions for the assumption to hold and explain why imposing L^2-regularization can ensure stationarity.

This paper will be more convincing if the above issues are addressed properly, and I will be happy to raise my score.

[1] Sebastian Ruder. An overview of gradient descent optimization algorithms. arXiv preprint arXiv: 1609.04747, 2017.

---

> ### Author Response · Authors · 2018-11-14
> **Adam+AMSGrad added; further optimization of adaptive scheduling hyperparameters performed; stationarity caveats clarified**
>
> Thank you so much for your very constructive comments! Please see below for our responses to your comments (and note our responses to other reviewers as well).
>
>
> (1) According to your suggestion, we have carried out two additional sets of experiments.
>
> (1-I) First, we carried out the experiments with Adam/AMSGrad [J. Reddi et al. (2018)] with the default hyperparameters. As we found AMSGrad to be marginally better than Adam for the task at hand (see newly added Appendix C if wished), we included the AMSGrad in the main text. Please see the figures 4(a) and 4(b) in the revised version. For the MNIST classification task with MLP, the proposed adaptive algorithm outperforms AMSGrad in terms of accuracy attained at long time and also has a quick convergence. For the CIFAR-10 classification task with CNN, the proposed adaptive algorithm again outperforms AMSGrad in terms of accuracy attained at long time, but its initial accuracy gain is visibly slower than AMSGrad.
>
> (1-II) Second, given the initial speed lag stipulated above for the CIFAR-10 task, we further carried out an extra hyperparameter search. We in turn found that a combination of the scheduling hyperparameters X and Y that outperforms AMSGrad in terms of long-time accuracy while also competing well in terms of initial accuracy gain (see Appendix C if wished). However, since it would be unfair to claim victory with the "task-specific-finely-tuned" adaptive scheduling against the "out-of-the-box" preset scheduling and AMSGrad, in the main text we still present the original results (+AMSGrad) with the "out-of-the-box" X and Y.
> These two additional experiments support the advantage of the adaptive method proposed herein, at least for the image classification tasks described in the draft. This is not to claim the universal applicability of the proposed method and further tests on the near state-of-the-art architectures on image classification tasks and also NLP tasks (for which Adam is often reported to be better than scheduled SGD) should be carried out.
>
>
> (2) The question of whether or not the SGD, viewed as a time-homogeneous Markov chain, holds stationary distributions is interesting, especially in the absence of the convexity assumption on loss-function landscapes. Our approach is empirical -- i.e., we prove and experimentally check various necessary conditions for stationarity, including the relation (FDR1) (c.f. Section 3.1) in addition to plateaus of loss and theta^2 (which we checked), and only then use its consequences [e.g. the relation (FDR2) to probe the landscape properties and (FDR1) to adaptively reduce the learning rate]. Figuring out under which conditions such stationary distributions exist is nontrivial. Specifically, if its formulation is too generic so as to include all the network architectures, then we would find a counterexample to such a putative existence statement. The case in point is a model with the cross entropy loss without regularizer, whose model parameters logarithmically diverge and never reach stationarity [please see footnote 2 of Neyshabur et al. (2014), page 4 of Neyshabur et al. (2017), and figure 1(B) of Soudry et al. (2017), which are all cited in the draft now]; there essentially is no local minimum with finite model parameters. That being said, this is the only class of nontrivial counterexamples that comes to our minds, and the said cascade can be avoided by the L^2 regularization.
>
> Given the comment, we nonetheless have extensively revised the Section 2.3.4 which can be interpreted as another paraphrased response to your comment.
>
>
> We believe that both of the two concerns have now been properly addressed, which in turn strengthened our results (especially due to the inclusion of Adam/AMSGrad algorithms). We thank you again for constructive comments and would appreciate further feedback if any.

---

### Official Review · AnonReviewer2 · 2018-11-02
**An innovative paper to assess equilibration in SGD**

**Rating:** 8
**Confidence:** 5

**Review:**

The paper introduces the concept of fluctuation-dissipation relations to stochastic gradient descent. These relations hold for certain observables in physical systems in equilibrium. In the context of SGD as a non-equilibrium process with a stationary density, they allow to quantify how far away this process is from its stationary state.

One of the strengths of the paper is that it works in the discrete-time formalism and uses the master equation, as opposed to other recent works that used the continuous-time limit of SGD to derive related (yet different) results. Furthermore, the formalism does not even rely on a locally quadratic approximation of the loss function, or on any Gaussian assumptions of the SGD noise. To the best of my knowledge, all of this is very innovative. Ultimately, the authors propose a practical algorithm to adaptively lowering the learning rate based on testing fluctuation-dissipation relations.

This is an interesting paper which I recommend to accept. It not only shows new theoretical results, but also conforms their validity in real-world experiments.

I have only a few questions / comments:

1. In Eq. 17 and others where the scalar product of theta and grad(f) occurs, is it implicitly assumed that the optimum of f is at theta=0?
2. In Fig. 2, the distinction between solid and dotted curves could be made better visible.
3. For completeness, it would be good to add the following citation:
Stephan Mandt, Matthew D. Hoffman, and David M. Blei. "Continuous-time limit of stochastic gradient descent revisited." NIPS 2015 Workshop on Optimization for Machine Learning.

---

> ### Author Response · Authors · 2018-11-14
> **Figure 2 revised; Mandt et al. (2015) cited.**
>
> Thank you very much for your generous review! Please see below for our responses to your comments (and note our responses to other reviewers as well, especially the discussion with the Reviewer 1 which led to additional experiments).
>
> (1) The short answer is that "no, it does not assume that the optimum is at theta=0; please see the comment after the equation (FDR1')." In more detail: as you note, the expression first looks translationally non-invariant, picking up the origin as a special point. However, due to <grad(f)>=0, the reference point of theta fluctuations can be shifted to anywhere one likes [i.e. replace theta*grad(f)->(theta-theta_{ref})*grad(f) with arbitrary choice of constant theta_{ref}] without affecting the stationary-state average value of O_L, as also mentioned in the footnote 4. (In practice, numerical accuracy/convergence speed to the stationary value can depend on a choice of the reference point.)
>
> (2) We agree that the distinction between solid and dotted curves were not so visible. After playing around with figure parameters to increase the contrast, we decided to reverse dotted and solid (now O_L is solid and O_R is dotted) and further made O_R thicker and colored lighter.
>
> (3) Thank you for bringing up a missing reference by Mandt et al.,  which we have now included along with their longer journal publication in 2017.

---

### Meta-Review · Area_Chair1 · 2018-12-13

**Confidence:** 5
**Recommendation:** Accept (Poster)

**Metareview:**

The paper presents interesting idea, but the reviewers ask for improving further paper clarity - that includes, but is not limited to, providing in-depth explanation of assumptions and also improving the writing that is too heavy and difficult to understand.

---

> ### Author Response · Authors · 2018-12-21
> **Metareply**
>
> Thank you for accepting the paper! We believe that these points have been fully addressed during the rebuttal period and that the revised paper provides in-depth explanation of assumptions, comes with stronger experimental results, and offers the writing that is hopefully easier to follow than the original.